# Evolution, persistence, and host adaption of a gonococcal AMR plasmid that emerged in the pre-antibiotic era

**Wearn-Xin Yee[1], Muhammad Yasir[2], A. Keith Turner[2], David J. Baker[2], Ana Cehovin[1]\*, Christoph M. Tang[1]\***

1 Sir William Dunn School of Pathology, University of Oxford, OXFORD, United Kingdom, 2 Quadram Institute, NORWICH, United Kingdom

\* ana.cehovin@path.ox.ac.uk (AC); christoph.tang@path.ox.ac.uk (CMT)

**Data Availability Statement:** All relevant data are within the manuscript and its Supporting Information files.

## Abstract

Plasmids are diverse extrachromosomal elements significantly that contribute to interspecies dissemination of antimicrobial resistance (AMR) genes. However, within clinically important bacteria, plasmids can exhibit unexpected narrow host ranges, a phenomenon that has scarcely been examined. Here we show that pConj is largely restricted to the human-specific pathogen, *Neisseria gonorrhoeae*. pConj can confer tetracycline resistance and is central to the dissemination of other AMR plasmids. We tracked pConj evolution from the pre-antibiotic era 80 years ago to the modern day and demonstrate that, aside from limited gene acquisition and loss events, pConj is remarkably conserved. Notably, pConj has remained prevalent in gonococcal populations despite cessation of tetracycline use, thereby demonstrating pConj adaptation to its host. Equally, pConj imposes no measurable fitness costs and is stably inherited by the gonococcus. Its maintenance depends on the co-operative activity of plasmid-encoded Toxin:Antitoxin (TA) and partitioning systems rather than host factors. An orphan VapD toxin encoded on pConj forms a split TA with antitoxins expressed from an ancestral co-resident plasmid or a horizontally-acquired chromosomal island, potentially explaining pConj's limited distribution. Finally, ciprofloxacin can induce loss of this highly stable plasmid, reflecting epidemiological evidence of transient reduction in pConj prevalence when fluoroquinolones were introduced to treat gonorrhoea.

## Author summary

Plasmids are extrachromosomal elements that disseminate antimicrobial resistance (AMR) among bacteria. *Neisseria gonorrhoeae* is a leading cause of sexually transmitted disease and a concern due to increasing AMR. It contains a restricted repertoire of plasmids, including a conjugative plasmid, pConj, which disseminates plasmid-mediated AMR. We show that, in contrast to broad host range plasmids, pConj is largely restricted to and adapted to *N. gonorrhoeae*, and has been remarkably conserved since it first emerged over 80 years ago. pConj is highly persistent, with no plasmid loss detected after 160 generations under standard laboratory conditions. We were unable to identify any

**Funding:** The work was supported by an A*STAR NSS PhD scholarship awarded to WXY and the Wellcome Trust (award number 214374/Z/18/Z to CMT). The funders had no role in study design, data collection and analysis, decision to publish, or preparation of the manuscript.

**Competing interests:** The authors have declared that no competing interests exist.

chromosomal gene to account for the success of pConj. Instead, the lack of fitness costs and co-operative effects of maintenance systems result in its stable inheritance. Of note, pConj harbours an orphan VapD toxin that can be neutralised by VapX antitoxins expressed by a co-resident plasmid; this potential 'split' toxin:antitoxin system allows exquisite association of pConj with the gonococcus. Finally, we show that ciprofloxacin can induce pConj loss, mirroring the reduction in pConj carriage in the gonococcal population following introduction of this antibiotic for gonorrhoea, and paving the way for approaches to eliminate plasmid-mediated AMR in this important human pathogen.

## Introduction

Plasmids are extrachromosomal elements which confer beneficial traits on bacteria including antimicrobial resistance (AMR) [1]. Plasmids can be transferred and maintained in different, unrelated species [2], and such broad host range plasmids have been extensively studied [2]. Narrow host range AMR plasmids are found in pathogenic bacteria including *Klebsiella* spp. and *Acinetobacter* spp. [3–5]; despite their clinical relevance, little is known about the mechanisms of their host restriction and maintenance.

Here we describe the evolution and maintenance of a narrow host plasmid in *Neisseria gonorrhoeae*. *N. gonorrhoeae* (the gonococcus) causes gonorrhoea, a serious threat to sexual and maternal health, and a co-factor for HIV infection [6]. The bacterium has evolved resistance against most available antibiotics so has been classified as a priority pathogen by the World Health Organisation and Centers for Disease Control and Prevention [7,8]. pCryp is an almost ubiquitous 4.2 kb plasmid of unknown function [9], while p*bla* (3.2–9.3 kb) and pConj (39–42 kb, markerless or carrying *tetM* from Tn*916*) led to the discontinuation of penicillin and tetracycline for treating gonococcal disease, respectively [8,10]. pConj is conjugative and disseminates itself and p*bla* [11–13] so is central to the spread of AMR. pConj could acquire further elements conferring resistance against other antibiotics including macrolides [14], while only one or two amino acid changes are needed for p*bla* to encode an extended spectrum beta-lactamase (ESBL) [15]. These changes would undermine currently recommended therapies against the gonococcus [16,17]. Therefore, it is crucial to understand how pConj is maintained in gonococcal populations, as this could inform approaches to combat AMR in this important human pathogen.

Plasmids deploy strategies to ensure their stable inheritance in bacteria. Partitioning systems segregate plasmids to the poles of dividing bacteria so that each daughter cell contains a plasmid following division [18]. In addition, Toxin:Antitoxin (TA) systems encode a toxin and a cognate antitoxin, and promote plasmid maintenance through post-segregational killing (PSK). In Type II TA systems, daughter cells which fail to inherit a plasmid are killed through the unopposed activity of the toxin once the protein antitoxin has been degraded [19].

pConj encodes a predicted partitioning system, consisting of a ParA ATPase and ParB DNA binding protein, while the genetic load (GL) region of the plasmid encodes two putative epsilon-zeta (ε:ζ) type II TA systems (Fig 1A) [20]. ε:ζ2 shares ~40% amino acid sequence identity with a characterised streptococcal ε:ζ system [21], while some gonococci carry ε:ζ*3* instead of ε:ζ*2* [9]. The GL region also encodes an uncharacterised orphan VapD toxin which lacks a cognate VapX antitoxin found in *vapXD* TA systems [22,23]. None of these systems have been characterised in the gonococcus, although ε:ζ1 is a functional TA system in *Escherichia coli* [21].

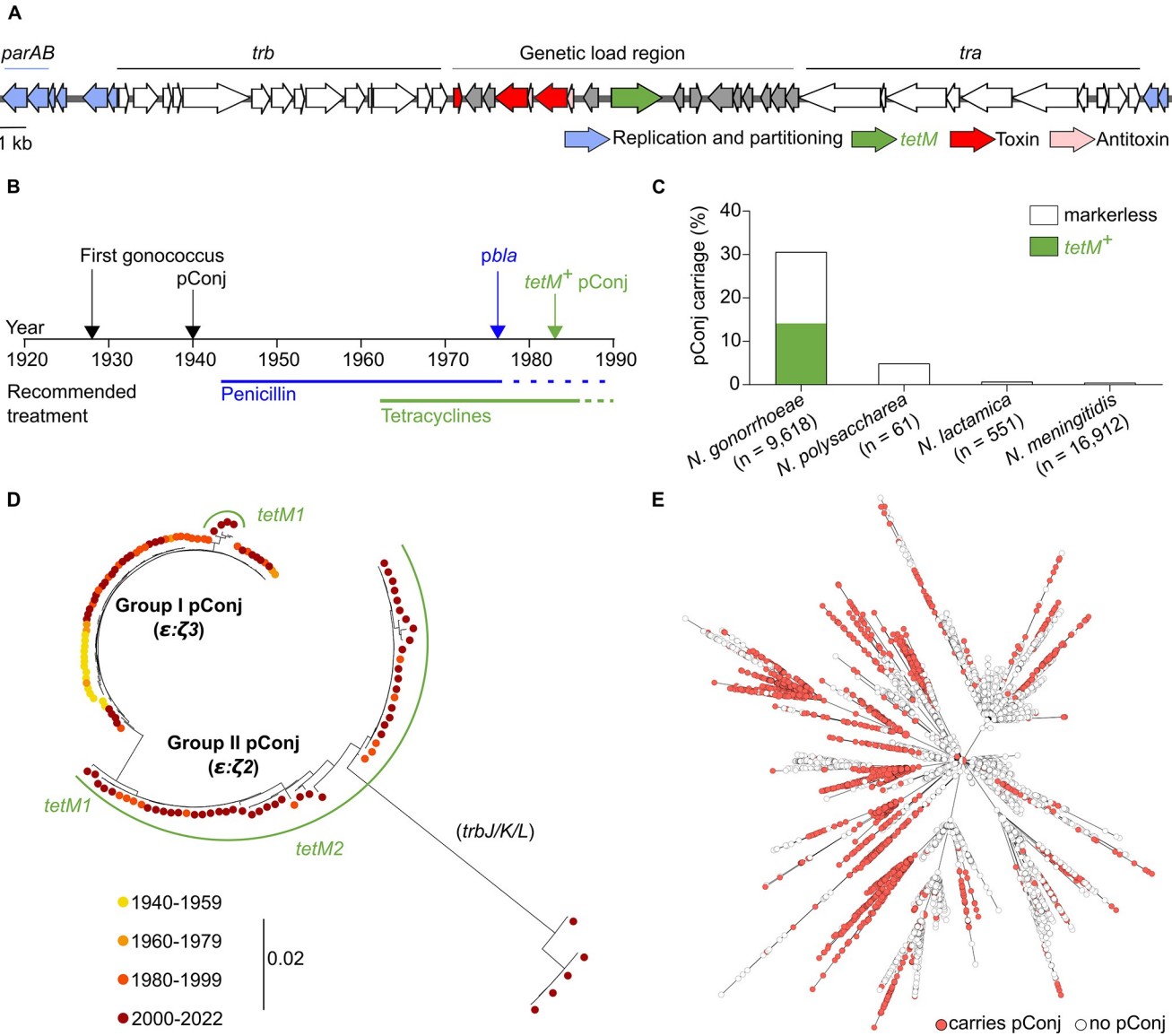

**Fig 1. pConj is conserved and largely restricted to the gonococcus.** (**A**) Map of pConj showing four regions involved in replication/partitioning (including *parAB*), mating bridge formation (*trb*), conjugation (*tra*) and the genetic load region which includes three toxin-antitoxin (TA) related loci. (**B**) Appearance of plasmids in *N. gonorrhoeae* based on WGS at PubMLST and previous reports; pConj first occurred in strains with pCryp. Years during which penicillin and tetracycline monotherapy were recommended in the USA are as shown. (**C**) pConj is present in four species of *Neisseria*; only species with ≥40 WGS are included. pConj was not detected in *N. bergeri*, *N. cinerea* and *N. subflava*. Only *N. gonorrhoeae* carried *tetM*⁺ pConj. (**D**) There are two phylogenetic groups of pConj, depending on the presence of ε:ζ2 or ε:ζ3 TA systems. Sequences were aligned independently of *tetM* and the surrounding transposon. Each dot represents an isolate, colour-coded according to year of isolation. (**E**) Despite pConj's conservation, the plasmid is found in multiple lineages across the gonococcal population. Each dot represents an isolate, colour-coded according to pConj carriage.

We show that pConj is largely restricted to the gonococcus and track its evolution from when it first appeared over 80 years ago to now. Even though *N. gonorrhoeae* is a highly variable pathogen [24] with pConj present in multiple lineages, pConj has been remarkably conserved over this time except for acquisition of *tetM* and TA systems. The plasmid does not impose obvious fitness costs and is stably inherited, explaining the persistence of pConj after tetracycline was discontinued for treating gonococcal infection. Transposon insertion sequencing (TIS) indicates that pConj does not rely on any non-essential chromosomal gene

for its maintenance. Instead, plasmid-encoded TA and partitioning systems act in concert to ensure pConj maintenance. Of note, we found that the orphan VapD encoded by pConj is part of a split TA system with VapX antitoxins encoded by other mobile genetic elements, pCryp or a horizontally-acquired Type IV secretion system (T4SS) genomic island, potentially explaining the restriction of pConj to *Neisseria* spp.. As the ParAB partitioning system is important for the vertical transmission of pConj, we tested whether ciprofloxacin, which can impair plasmid segregation through its effect on DNA gyrase [25], can be used to cure pConj. Importantly, exposure to ciprofloxacin enhances pConj loss from the gonococcus. This is consistent with the decrease in pConj prevalence that occurred when this antibiotic was introduced for treating gonococcal disease [26,27], and provides proof-in-principle that this highly stable and adapted plasmid can be eliminated from *N. gonorrhoeae*.

## Results

### pConj is conserved, restricted to *Neisseria* spp. and does not impose fitness costs

To investigate the evolution and restriction of pConj, we initially examined whole genome sequences (WGS) of gonococcal isolates deposited in PubMLST (n = 9,618, S1 Table) for the presence of pConj, pCryp and p*bla*. The earliest sequenced gonococcal isolate dates to 1928 and carries pCryp (isolate DO371). Subsequently, markerless pConj (*i.e.* lacking *tetM*) first appeared in 1940 (isolate DO12954) in isolates with pCryp. $tetM^+$ pConj was described later in 1983 [28]; shortly after, tetracycline monotherapy ceased in the USA [7] (Fig 1B). The first isolate with p*bla* (isolate 215/-02) dates from 1979 and harbours pConj and pCryp; this coincided with detection of p*bla*-carrying isolates in the USA in 1976 [29] (Fig 1B). Spread of p*bla* led to the discontinuation of penicillin treatment for gonorrhoea [8].

We found that pConj is only found in *Neisseria* spp., and largely restricted to *N. gonorrhoeae* (in 30.7% of isolates, 2,952/9,618). The plasmid is sporadically found in *Neisseria meningitidis* (0.5% of isolates, 81/16,912), *Neisseria lactamica* (0.7% of isolates, 4/551), and *Neisseria polysaccharea* (4.9% of isolates, 3/61), and absent from other *Neisseria* spp. (Fig 1C and S1 Table). Therefore, unlike many AMR plasmids, pConj is restricted to a few closely related species, and overwhelmingly associated with a single species, *N. gonorrhoeae*.

As pConj has existed in *N. gonorrhoeae* for over 80 years including the pre-antibiotic era, we next examined its evolution by analysing the sequences of 126 representative plasmids from all previously described seven pConj variants [9] in strains from different lineages (S2 Table). Phylogenetic analysis shows that there are two distinct pConj groups, I and II. Group I pConj (comprising pConj variants 4 to 7 [9]) appeared first, carries an ε:ζ3 TA system, and is markerless or $tetM^+$, while group II pConj comprises variants 1 to 3 which acquired ε:ζ2 and *tetM* (Fig 1D).

ζ2 and ζ3 share 50–70% amino acid similarity and retain predicted active site residues (S1 Fig). Apart from *tetM* and ε:ζ, pConj has remained remarkably conserved: group I pConj from the 2000s are virtually indistinguishable from plasmids circulating in the 1940s (Fig 1D). An exception is a sub-set of recent group II *N. gonorrhoeae* pConj with mutations in genes involved in mating pair formation (Fig 1D). The conservation of gonococcal pConj contrasts with its diversity in *N. meningitidis* (S2 Fig). Despite its conservation, pConj is widely disseminated across multiple lineages within the gonococcal population (Fig 1E), consistent with previous reports [9], suggesting that pConj has remained conserved despite evolutionary pressure to the gonococcal chromosome.

pConj might be widespread in *N. gonorrhoeae* because of the use of tetracyclines to treat gonococcal infection [30]. To examine this, we determined the prevalence of pConj in

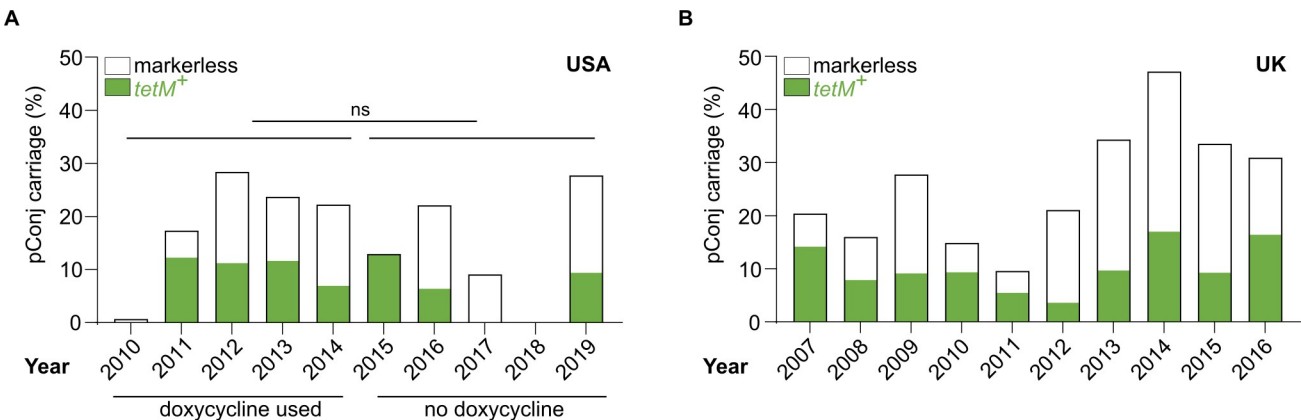

**Fig 2. pConj has persisted in the absence of tetracycline.** Prevalence of pConj in WGS of isolates from (**A**) USA and (**B**) UK. Doxycycline was recommended as part of a dual therapy for treatment before 2014 in USA; pConj carriage (including markerless plasmids) remained constant during/after doxycycline treatment (n = 2,517 isolates), as determined by unpaired *t*-test. pConj has remained prevalent in the UK after tetracycline treatment (2007–2016, n = 3,240 isolates). ns, $p > 0.05$.

gonococci isolated in the USA when tetracyclines were recommended as part of a dual therapy for treating gonococcal disease [30] (2010–2014, n = 1,467) and compared this with subsequent years when tetracyclines had been discontinued [31] (2015–2019, n = 1,050, S3 Table). Of note, the prevalence of *tetM*⁺ pConj did not fall following the cessation of tetracycline use ($p = 0.4547$, unpaired *t*-test Fig 2A). Furthermore, the prevalence of strains harbouring markerless pConj has not changed over this time ($p = 0.7846$, Fig 2A). Therefore, pConj has persisted in the absence of selection imposed by tetracycline treatment for gonococcal disease. Similarly, pConj carriage has been steady in the UK since 2007 (n = 3,240, Fig 2B and S3 Table), even though tetracyclines have not been used to treat gonorrhoea in this country since 2004 [32].

The persistence of pConj in gonococcal populations is mirrored by its stable vertical transmission and lack of fitness costs. To determine the loss of pConj, group I and II pConj containing *tetM* (variants 4 and 1, respectively) were introduced into *N. gonorrhoeae* FA1090 and tagged with *gfp:kan* (generating pC1^GFP and pC4^GFP for variants 1 and 4, respectively) to allow detection of pConj loss at low frequency (*i.e.* ≤ 0.3%). Introduction of *gfp:kan* did not affect the growth of strains (S3 Fig). No loss of pC1^GFP or pC4^GFP was observed over 160 generations in the absence of selection (S3 Fig). pC1^GFP was used in all subsequent experiments as variant 1 pConj is common in gonococci [9]. There was no difference in the growth of FA1090 +/- pConj in fastidious broth (FB) or gonococcal base media (GCBL, S3 Fig, two-way ANOVA with Sidak's multiple comparisons, $p > 0.05$). Therefore, pConj does not impose detectable fitness costs on the gonococcus, even though it accounts for approximately 10% of the genetic content of strains (the plasmid to chromosome copy number is approximately 4.6, S3 Fig). Taken together, pConj is a highly conserved plasmid that is largely restricted and adapted to the gonococcus in which it is stably inherited.

## TIS does not reveal host genes for pConj maintenance

The adaptation and restriction of pConj to the gonococcus led us to hypothesise that the plasmid relies on host-specific genes for its maintenance. To identify chromosomal genes involved in pConj maintenance, we constructed a library of FA1090 mutants by *in vitro* transposon mutagenesis of FA1090 chromosomal DNA followed by uptake of mutagenised DNA *via* transformation. pConj was then introduced into the library. The resulting library containing

>100,000 unique insertion sites (UIS) was grown with or without tetracycline in duplicate cultures for 56 generations; transposon insertion sites were determined using the Tradis-*Xpress* nucleotide sequencing method [33] after the 8th, 16th and 56th generation. The number of UIS diminished over time, with 183,919 UIS in the initial library, 81,655–89,336 UIS by the 8th generation and 54,708–70,973 UIS by the 16th generation. By the 56th generation, there were only between 5,815 and 6,442 UIS in the cultures, too few to identify genes involved in plasmid maintenance (S1 Dataset). The profile of insertions was highly consistent between cultures (S4 Fig). No chromosomal genes were found to contribute to plasmid maintenance after eight generations (S4 Table). Significant hits (*i.e.* enriched in cultures lacking tetracycline, log fold change < 0, $q < 0.01$) were only identified after 16 generations (Fig 3 and S4 Table).

However, all hits were due to differences in one or two insertions per gene (Fig 3), which is characteristic of false positive hits [34]. To determine whether they were false positives, mutants Δ*neis1066*, Δ*neis1845*, Δ*neis2592*, which gave the most significant *q*-values or highest counts per million (logCPM) identified by Tn-Seq were constructed in pC1^GFP. Mutants were passaged for 16 generations, and plasmid loss was determined by detecting the absence of GFP. No plasmid loss was observed (LOD = 0.3%), confirming that TIS yielded only false positive hits.

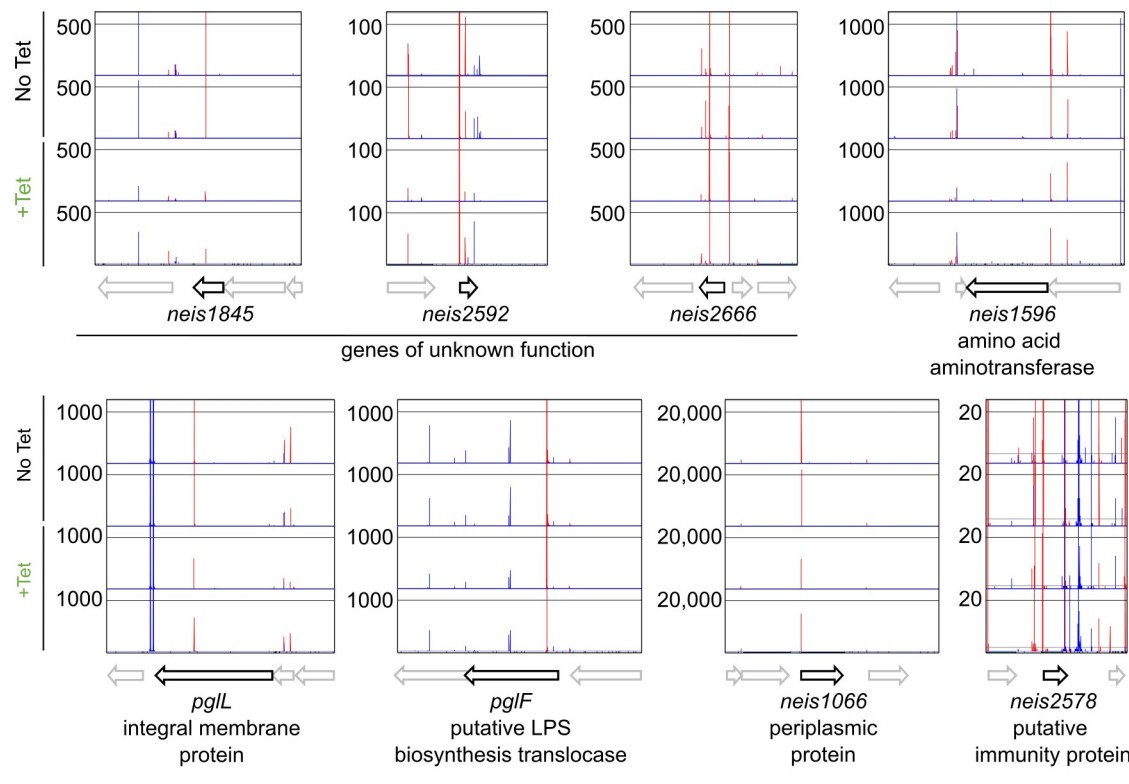

**Fig 3. TIS does not identify chromosomal genes involved in pConj maintenance.** Transposon insertion sites of significant hits at the 16th generation (*q* < 0.01, Bio-Tradis; log fold change < 0) visualised in Artemis; gene orientation is shown. Graphs show the distribution, numbers and orientation of transposon insertion mutant library obtained under both control and tetracycline conditions (two independent biological repeats). Each vertical line indicates a UIS, with the height reflecting the number of mutants at each site. Red and blue lines indicate transposon insertions in the forward and reverse direction, respectively. Blue lines are plotted in front of red, masking some red insertions. The maximum number of insertions displayed in each panel is marked individually.

Therefore, after 16 generations we also inspected the distribution of transposon insertions in/around 16 genes encoding proteases and DNA replication machinery, which influence plasmid maintenance in other bacteria [35–38]. No differences were detected in the distribution or frequency of insertions around these genes (S4 Fig). Overall, these results suggest that pConj does not rely on a single non-essential chromosomal gene for its stable inheritance.

## TA systems co-operate to maintain pConj

As we did not identify any host genes contributing to pConj stability, we next examined plasmid genes. Initially, we determined whether the TA systems encoded by the GL region contribute to pConj stability. We replaced the entire GL region in pConj (*neis2203* to *vapD* inclusive [20], Fig 4A) with *gfp*:*kan*, generating pCΔGL, and assessed plasmid loss after growing bacteria for 30–40 generations under non-selective conditions. Under these conditions, loss of pC1$^{GFP}$ (pConj with the GL region) was below the limit of detection while deletion of the GL region reduced pConj stability (loss of pCΔGL, 8.5% ±3.1, Fig 4B).

To define which TA systems encoded in the GL region contribute to plasmid stability, we next deleted or inactivated the toxins individually (pCtΔ*vapD*, pCt:ζ1$^{K115A}$, pCtΔζ2) or in combination (pCtΔ*vapD*:ζ1 and pCtΔTA, Fig 4C). *trbC*, which encodes the conjugative pilus, was removed from all plasmids to prevent pConj re-acquisition by conjugation [39]. No pConj loss was detected after growth for 30–40 generations following deletion of *vapD* or ζ2. However, inactivation of ζ1 led to low level plasmid loss (0.33% ±0.3) demonstrating that this TA system contributes to pConj maintenance (Fig 4C).

Subsequent removal of *vapD* then ζ2 from pCtΔζ1 mutant led to stepwise increases in pConj loss (loss of pCtΔ*vapD*:ζ1 and pCtΔTA, 3.2% ±0.4 and 5.6% ±1.6, respectively, $p = 0.041$ and

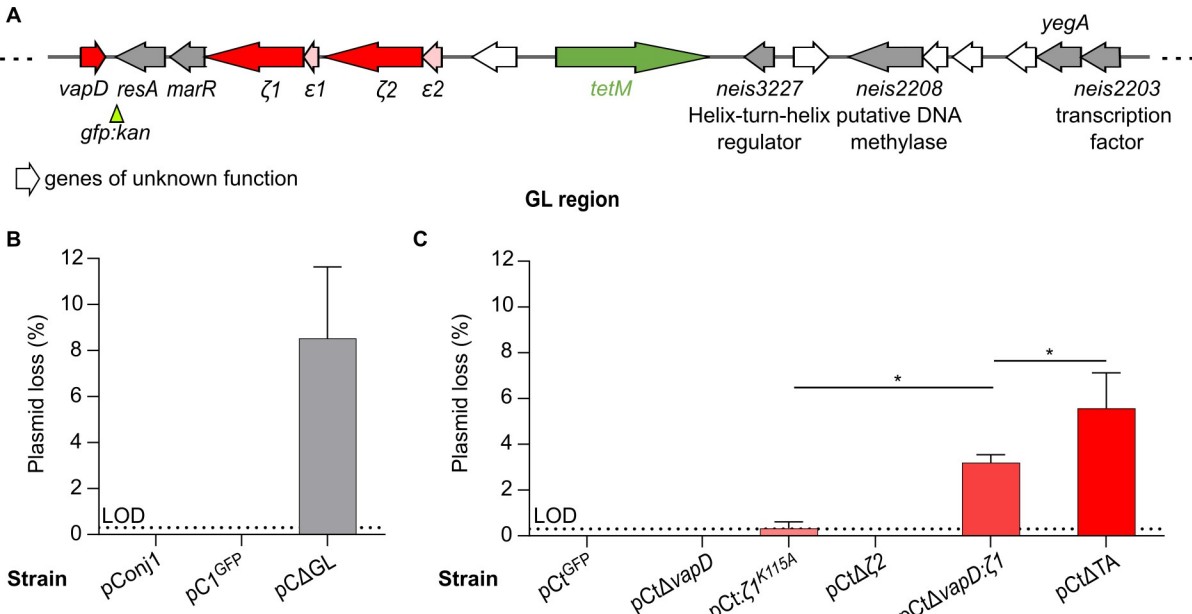

**Fig 4. TA systems cooperate to maintain pConj.** (**A**) Schematic of the genetic load region. *gfp*:*kan* was used to monitor the presence of pConj. Genes and related functions are annotated based on PubMLST. (**B**) Replacement of the genetic load region with *gfp*:*kan* (pCΔGL) resulted in detectable plasmid loss. (**C**) pConj lacking individual TA systems (pCtΔ*vapD*, pCt:ζ1$^{K115A}$, pCtΔζ2) were stably maintained, whereas loss of combinations of TA systems (pCtΔ*vapD*:ζ1 and pCtΔTA) resulted in plasmid loss within 40 generations (LOD = 0.3%). All assays consist of three independent repeats and results were analysed with one-way ANOVA with Sidak's multiple comparisons, shown as mean ± SD; * $p \leq 0.05$).

$p = 0.019$, one-way ANOVA with Sidak's multiple comparisons test, Fig 4C) indicating that *vapD* and *ε:ζ2* also support the maintenance of pConj in the absence of other TA systems. Overall, our results demonstrate that pConj TA systems act redundantly to maintain the plasmid.

## Interactions between mobile genetic elements involved in pConj maintenance

To understand the mechanism by which VapD expressed by pConj (VapD[pConj]) contributes to plasmid maintenance, we examined whether VapD[pConj] is toxic. VapD[pConj] shares homology with *Helicobacter pylori* [40] and *Haemophilus influenzae* [22] VapD (34.8% and 34.4% amino acid similarity respectively), and is also predicted to be a dimer by AlphaFold (Fig 5A). VapD[pConj] expression in *E. coli* under an arabinose-inducible promoter [41] led to a marked reduction in bacterial survival ($p < 0.0001$, two-way ANOVA with Tukey's multiple comparisons test, Fig 5B), demonstrating that VapD[pConj] is toxic. We next investigated if VapD[pConj] interacts with VapX homologues encoded elsewhere in the gonococcal genome. Of note, pCryp, which was in the first gonococcal isolate found with pConj, encodes a *vapXD* TA system (Fig 1B). pCryp, similar to pConj, is restricted to *Neisseria spp.*, and is most common in *N. gonorrhoeae* (96.8% of isolates, 9,313/9,618, S1 Table). VapXD is also associated with a horizontally acquired chromosomal island containing a Type IV secretion system (T4SS) in some gonococci [9]. Due to the high prevalence of pCryp [9,42], we tested whether expression of *vapX[pCryp]* can prevent VapD[pConj] toxicity. Results demonstrate that VapX[pCryp] abrogates VapD[pConj] toxicity ($p = 0.0069$, Fig 5B), consistent with VapD[pConj] forming a split TA system with VapX[pCryp].

We also examined the association between pConj with VapX and found that pConj carriage is significantly higher in strains harbouring *vapX* (pConj in 31.2%, 2,914/9,316 of *vapX*-carrying strains) than in gonococci which do not carry *vapX* (12.5%, 38/302, $p < 0.0001$, Fisher's exact test Fig 5C). Thus, *vapX* carrying strains are also more likely to carry pConj (odds ratio 3.1, 95% confidence interval 2.2–4.4, Woolf logit).

We hypothesised that the association between pConj VapD and pCryp VapX might also be evident in other *Neisseria spp.* that carry pConj. Analysis of WGS suggested that *N. cinerea*, *N. subflava* and *N. bergeri*, which did not carry pConj (Fig 1C), had very low levels of *vapX* carriage (0/40 and 0/53 for *N. cinerea* and *N. subflava* respectively, and 1/60 for *N. bergeri*). For pConj-carrying species, although no isolates carrying *vapX* was identified for *N. polysaccharea* (0/61), *vapX* carriage in *N. lactamica* was 68.4% (377/551) and 1.1% (194/16,912) in *N. meningitidis*; *vapX* in both species are not found on pCryp. Specifically, *vapX* in *N. meningitidis* is usually found on a horizontally-acquired island encoding a Type IV secretion system [43] (T4SS, in 87.1%, or 169/194 of *vapX*-carrying strains), and less commonly on pCryp (10.3%, 20/194). The *vapX* allele on the T4SS was designated *vapX[Nm]*, and differs from *vapX[pCryp]* by an insertion of three amino acid residues near the N-terminus. We confirmed that VapX[Nm] also neutralises VapD[pConj] following expression in *E. coli* ($p = 0.0013$, Fig 5D). Furthermore, similar to *N. gonorrhoeae*, pConj carriage is significantly more frequent in meningococci harbouring *vapX*; pConj is present in 4.1% of meningococci with *vapX* (8/194 isolates) but only in 0.44% of strains lacking *vapX* (73/6,718 isolates, $p < 0.0001$, Fig 5E, odds ratio 9.8, 95% confidence interval 4.7–20, Baptista-Pike). Interestingly, *vapD* has been lost from 13 pConj-carrying meningococcal strains, and replaced by a 433 bp sequence within pConj (isolates 4313, 26256, 49535, 56700, 57863, 57509, 59291, 83101, 86009, 92506, 93290, 94878, and 116658; S1 Table). All pConj lacking *vapD* are found in *vapX*-negative strains. In contrast, every other pConj in *N. meningitidis* and *N. gonorrhoeae* contains *vapD*. Overall, these data indicate that VapX, encoded by pCryp or a chromosomal T4SS island, can form a split TA system with the orphan VapD expressed by pConj.

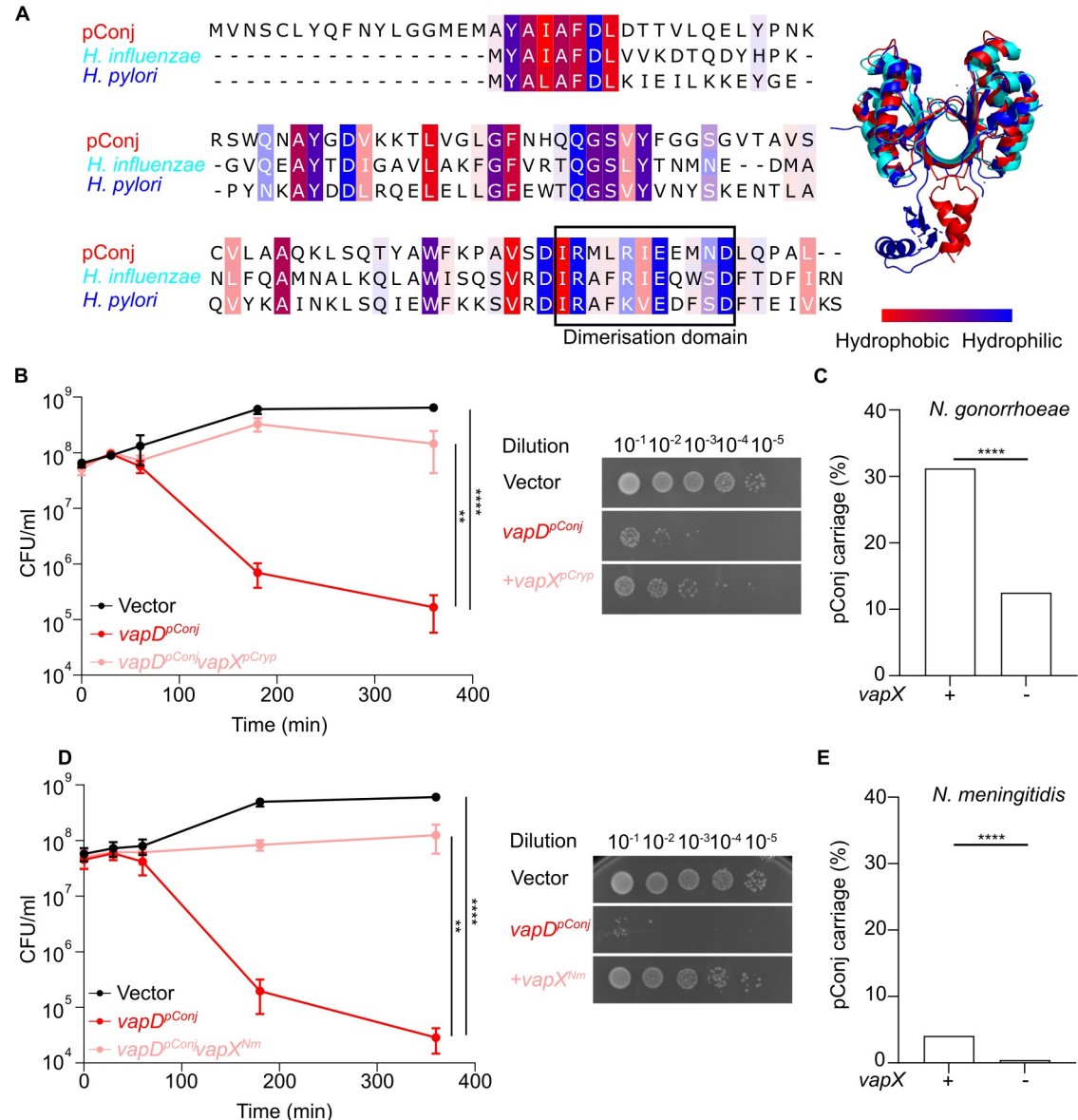

**Fig 5. Interaction between mobile genetic elements is associated with pConj carriage.** (**A**) VapD$^{pConj}$ is homologous to other VapDs. Shading intensity is proportionate to percentage conservation. AlphaFold predicted structure of VapD$^{pConj}$ (red) has similar architecture to VapD from *H. pylori* (PDB 3UI3, dark blue) and *H. influenzae* (PDB 6ZN8, light blue). (**B**) VapD$^{pConj}$ is toxic in *E. coli*, and is neutralised by VapX$^{pCryp}$. Viability of *E. coli* (CFU/ml) after induction with 0.2% of L-arabinose is shown. (**C**) pConj carriage in *N. gonorrhoeae* with *vapX* (n = 9,316) compared to isolates without *vapX* (n = 302) (**D**) VapD$^{pConj}$ toxicity in *E. coli* is neutralised by VapX$^{Nm}$. (**E**) pConj carriage in *N. meningitidis* isolates with *vapX* (n = 194) compared to isolates without *vapX* (n = 16,718). Toxin-antitoxin assays were carried out in three independent repeats, analysed with two-way ANOVA with Tukey's multiple comparisons and shown as mean ± SD. In **B** and **D** the number of CFU at 360 min post-induction are shown. pConj carriage was analysed with Fisher's exact test. ** $p \leq 0.01$, **** $p \leq 0.0001$.

## Interplay between TA and partitioning systems stably maintain pConj

Apart from TA systems, partitioning systems can promote the maintenance of low copy number plasmids. We therefore examined the effect of the putative ParAB partitioning system on pConj maintenance (Fig 1A). We generated pConj lacking *parB* (pCtΔ*parB*), and found that after only 6–9 generations approximately 15% of bacteria had lost the plasmid (Fig 6A). We

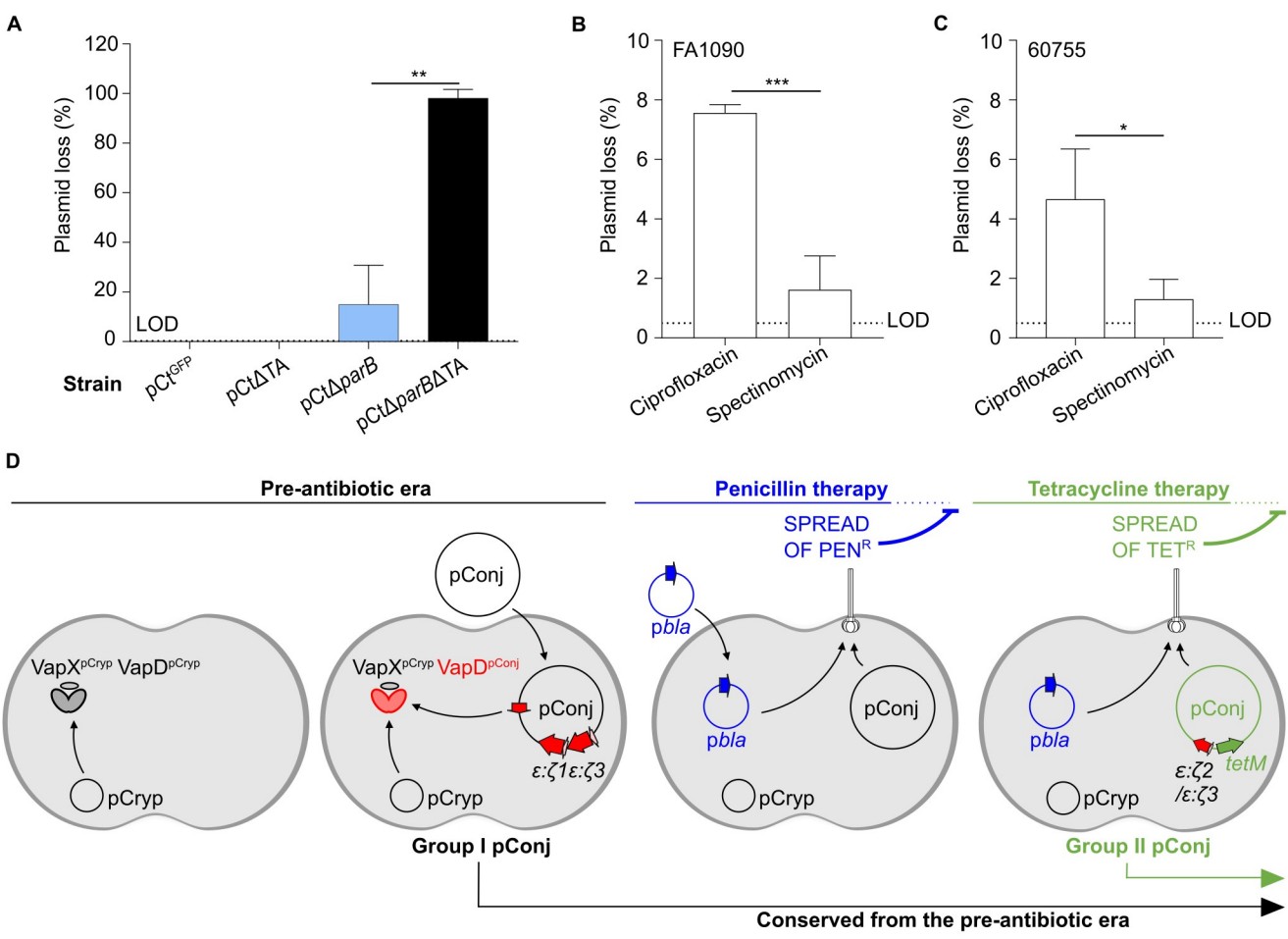

**Fig 6. pConj is stably maintained by multiple mechanisms.** (**A**) pConj is rapidly lost from strains lacking TA and/or partitioning systems (*i.e.* within 6–9 generations). Assays were carried out three times and analysed with Mann-Whitney test, LOD = 0.3%, shown as mean ± SD. ** $p \leq 0.01$. Treatment of (**B**) FA1090 or (**C**) 60755 with ciprofloxacin promotes pConj loss. Bacteria were grown overnight on GCB before incubation in 0.5 MIC of ciprofloxacin or spectinomycin. Colonies ($n \geq 200$; LOD = 0.5%) were tested for pConj carriage. Assays were carried out three times and analysed with student's unpaired *t*-test, shown as mean ± SD. * $p \leq 0.05$, *** $p \leq 0.001$. (**D**) As pConj enters *N. gonorrhoeae*, interactions between pCryp and pConj via VapXD contribute to the stabilisation of pConj in the gonococcus as well as its maintenance. pConj encodes TA and partitioning systems–therefore, pConj has developed multiple fail-safe mechanisms to ensure it is maintained in *N. gonorrhoeae*. While pConj is stably maintained in *N. gonorrhoeae*, antibiotic use facilitated spread of both itself and p*bla*, resulting in the spread of plasmid-mediated antimicrobial resistance. Currently, despite the lack of tetracycline use, pConj persists in *N. gonorrhoeae*; as such, treatment regimens no longer suggest the use of penicillin or tetracycline.

reasoned that the dramatic impact of ParAB might be masked by the activity of TAs, so additionally inactivated the TA systems in this plasmid (generating pCtΔ*parB*ΔTA). This led to almost all bacteria losing pConj over only 6–9 generations (98% ±3.6, $p$ = 0.0022 compared with pCtΔ*parB*, Mann-Whitney test, Fig 6A), demonstrating that pConj TA and partitioning systems operate in concert to prevent plasmid loss.

## Ciprofloxacin can promote pConj loss

Given the importance of the ParAB partitioning system, we attempted to eliminate pConj from the gonococcus by exposing bacteria to ciprofloxacin which can affect plasmid segregation by targeting DNA gyrase and/or topoisomerase [44]; spectinomycin was included as a control [45], and pCΔGL was used in these experiments to exclude the effects of TA systems.

We exposed bacteria to 0.5 Minimal Inhibitory Concentration (MIC) of ciprofloxacin or spectinomycin for 48 hours and assessed plasmid loss from FA1090 and 60755, an isolate from Kenya that naturally carries pConj; bacterial survival was similar in the presence of either antibiotics (S5 Fig). Compared with spectinomycin, exposure to ciprofloxacin resulted in significant pConj loss from both strains (for FA1090, 7.6% ±0.3 for ciprofloxacin *vs.* 1.6% ±1.1 for spectinomycin, $p = 0.0009$; for 60755, 4.7% ±1.7 *vs.* 1.3% ±0.7, $p = 0.03$ unpaired *t*-test, Fig 6B and 6C). These data demonstrate that ciprofloxacin, and other related antibiotics, could eliminate pConj from the gonococcus.

## Discussion

Gonorrhoea is a major public health problem due to increasing AMR [7], and *N. gonorrhoeae* carries multiple plasmids with limited host range, including pConj which contributes to the spread of high-level beta-lactam and tetracycline resistance [11]. Here, we show that pConj arose decades earlier than previously appreciated [46], is prevalent, remarkably conserved and imposes no detectable fitness cost on the gonococcus, despite accounting for around 10% of total genetic content. The stable inheritance of pConj relies on redundant and co-operative effects of partitioning and TA systems. The orphan VapD toxin on pConj forms a split TA system with antitoxins encoded on other mobile genetic elements, which may explain the plasmid's restriction to the gonococcus. Finally, we used ciprofloxacin, which interferes with partitioning, to provide a proof-in-principle that this highly adapted, stable plasmid can be cured through understanding mechanisms underlying its inheritance.

Toxicity and antitoxicity assays and epidemiological evidence indicate that pConj and pCryp form a stable association likely due to the interaction between $VapD^{pConj}$ and VapX. This is consistent with the initial appearance of pConj over 80 years ago in a gonococcal isolate with pCryp, and the observation that introduction of pCryp *vapX* into *E. coli* increases its ability to acquire pConj by conjugation [20]. The genetic location of *vapX* may be important, as both *N. lactamica* and *N. meningitidis*, which have low pConj carriage, do not carry *vapX* on the high-copy-number plasmid pCryp found in *N. gonorrhoeae* [47]. Furthermore, there could be limited exposure of both *N. meningitidis* and *N. lactamica* to pConj as they occupy different niches to *N. gonorrhoeae* [48]. The relationship between $VapD^{pConj}$ and $VapX^{pCryp}$ offers a potential explanation for the restricted distribution of pConj to *Neisseria* spp., even though its P1 replicon is typical of broad host-range plasmids [49].

Since its acquisition by the gonococcus, pConj has been remarkably conserved, unlike most other AMR plasmids [50,51]; major changes in pConj are limited to acquisition of *tetM*, $\varepsilon{:}\zeta$ and *trbJ/K/L*, which contrasts with how the gonococcal genome has evolved rapidly under antibiotic pressure [24]. *tetM* was previously shown to be acquired from a streptococcal transposon Tn*916* [52] and was suggested to be acquired independently twice, resulting in two *tetM* alleles [53]. Both group I and II pConj are prevalent and inherited stably, suggesting that both $\varepsilon{:}\zeta2$ or $\varepsilon{:}\zeta3$ promote pConj maintenance; differences in their activity is under further investigation. We also identified the sub-group in group II pConj as the lineage-specific pConj variant 2 previously characterised [9] and hypothesised that the extensive mutations could be due to horizontal acquisition or lineage-specific evolution of *trb* genes. Regardless, the long-term success of pConj is evident from its continued prevalence after tetracycline treatment for gonococcal infection ceased. Although gonococci could still be exposed to tetracyclines given to treat co-infections with *Chlamydia trachomatis* [54], this does not explain the persistence of markerless pConj. Of note, all analyses of pConj prevalence and emergence are dependent on collections and datasets available, thus increased sampling and collection from clinics in the future is paramount to understanding pConj spread and maintenance in the future.

Host genes can influence the fitness cost of plasmids [55], and plasmid-host co-evolution often results in plasmid stabilisation [56]. Our TIS analysis did not reveal any chromosomal factor required for pConj maintenance even after inspection of genes implicated in plasmid replication and segregation in other species [35–38]. We confirmed that this is unlikely due to reacquisition of pConj during liquid growth, as conjugation is low during liquid culture (conjugation rate, 2.6 x10$^{-3}$ over 24 hr). However, TIS cannot assess the role of essential genes or the concerted effect of multiple genes. Additionally, the number of UIS decreased over time, limiting the ability of TIS to detect significant hits in later generations. Instead, we found that the plasmid possesses TA and partitioning system for its stable inheritance. This has significant implications for understanding the persistence of low copy plasmids which express both TA and partitioning systems; plasmids such as pConj are segregated by partitioning systems during cell division, with the TA systems acting as a fail-safe mechanism should segregation fail. In this way, pConj appears to be entirely self-sufficient for its high prevalence in the gonococcus.

As our findings highlighted the importance of ParAB in pConj stability, we reasoned that the plasmid could be eliminated by drugs such as ciprofloxacin which interfere with DNA segregation [57]. We were able to eliminate pConj in a proportion of bacteria by exposing them to sub-lethal concentrations of ciprofloxacin. These findings mirror the observation that pConj carriage decreased following the emergence of ciprofloxacin resistance due to its introduction to treat gonorrhoea [26,27]. Furthermore, it provides proof in principle that molecules targeting TA/partitioning, including next-generation gyrase inhibitors, could be used to eliminate pConj from the gonococcus.

Overall, we show how plasmid-mediated resistance has evolved and is stably inherited in the high priority pathogen *N. gonorrhoeae* (Fig 6D). We found previously that *tetM*$^+$ pConj is more frequently found in LMICs than in wealthier countries, probably due to extensive and inappropriate use of antibiotics [9]. However, stopping tetracycline treatment has not reduced pConj prevalence, highlighting the exquisite adaptation of this plasmid to the gonococcus. Our work also provides a further example that AMR might not be eliminated by simply reducing antibiotic use. However, we show that it might be possible in the future to eliminate pConj, the main driver of gonococcal plasmid-mediated AMR, by understanding then targeting plasmid maintenance systems.

## Materials and methods

### Bacterial strains and growth

The strains and plasmids used in this study are listed in S5 and S6 Tables. *N. gonorrhoeae* was grown on gonococcal base media (GCB) supplemented with 1% v/v Vitox (Oxoid) and 1.1% w/v Bacteriological Agar No. 1 (Oxoid) [58]. For liquid growth, *N. gonorrhoeae* was grown in liquid GCB (GCBL) or fastidious broth (FB) with 1% Vitox, and incubated at 37°C in 5% $CO_2$ with shaking at 180–200 rpm. For FB, after autoclaving, the media was supplemented with sterile-filtered 35.9 μM pyridoxal (Sigma), 0.05% v/v polysorbate 80 (VWR International), 22.6 μM NAD (Sigma) [59] and 1% Vitox. *E. coli* was grown on Luria-Bertani Agar (LB, Oxoid) or in liquid LB with shaking at 180 rpm at 37°C. Supplements were added as needed at the following concentrations: for *E. coli*, 50 μg/ml kanamycin (Sigma), 20 μg/ml chloramphenicol (Sigma), and 40 μg/ml X-gal. For *N. gonorrhoeae*, kanamycin, tetracycline and erythromycin (all from Sigma) were used at 50 μg/ml, 2 μg/ml and 2 μg/ml, respectively; media for counter selection contained 6.4% w/v 2-deoxygalactose (Sigma).

### Gonococcal strain construction

Primers used in this study are listed in S6 Table. In general, gene deletion was achieved by amplifying approximately 1 kb up- and downstream of the target gene and fused with an

intervening marker for selection and a flanking vector by Gibson assembly (New England Bio-labs). The assembly reaction was then transformed into *E. coli* DH5α for selection and sequencing, before transformation into *E. coli*.

The *gfp:kan* cassette was generated from 280 bp of the *opaB* promoter [60], *sfGFP* [41], and the kanamycin resistance cassette *aph(3)-Ia* [39]. The insertion of the *gfp:kan* cassette included a duplication of the intergenic region between *vapD* and *res* which contains a predicted terminator to flank the cassette. The ζ1 K115A substitution [21] was introduced by site-directed mutagenesis of AAA to GCC at nt. 343–345 of *ζ1*; fragments up- and downstream of *ζ1*[K115A] were amplified from pC[GFP] using primers 259/284 and 253/283, respectively.

Constructs were amplified from plasmids with relevant primers (S6 Table) before transformation into gonococci; between 100 ng and 1000 ng of each linear construct was spotted to solid media and left to dry. Bacteria were streaked over the DNA or water control dry spots and incubated for 8–10 hours at 37˚C in 5% $CO_2$. Transformants were selected by plating on solid media containing appropriate antibiotics, and confirmed by sequencing.

For construction of FA1090 carrying wildtype pConj variant 1 or 4, FA1090 was conjugated with strains 60755 or 55496 respectively for 24 hours, before selection on tetracycline as previously described [39]. Individual colonies were then picked, and the *porB* gene was sequenced for strain confirmation.

## Plasmid stability and fitness costs

For plasmid stability assays, gonococcal strains were grown overnight on solid media and used to inoculate 15 ml FB at an $OD_{600}$ 0.001. Cultures were passaged every 24 hours, starting at $OD_{600}$ 0.001; approximately 300 individual colonies were plated every two passages (20 generations) to assess pConj loss using a UV transilluminator to visualise fluorescence (Invitrogen), resulting in a detection limit of 0.3%. After 160 generations, 40 individual colonies were re-streaked on media with or without tetracycline to confirm pConj carriage.

For assessing fitness cost, strains were grown overnight on solid media then used to inoculate GCBL at an $OD_{600}$ 0.1 in 96 well plates. The $OD_{600}$ was measured every 20 min over 8 hours using a FLUOstar Omega Microplate Reader (BMG Labtech), and plated out at 8 hours; 40 individual colonies were re-streaked on media with or without tetracycline to determine pConj carriage. Separately, for assays > 8 hours, strains were inoculated into 15 ml FB at a concentration of 5 x 10⁴ CFU/ml and sampled over 24 hours to follow bacterial viability.

## Plasmid copy number

To generate standards for assessment of pConj copy number, a 160 bp fragment of chromosomal *recA* was amplified with Q7 and Q8, and primers 121 and 119 were used to amplify a 965 bp fragment containing *res* from pConj. These fragments were cloned into pCR 2.1-TOPO (Thermofisher), which was used as a control plasmid. Plasmid concentration was measured using Qubit, serially diluted and added to qPCR mastermix for qPCR. Primers Q7/Q8 for *recA* and Q21/22 for *res* were used to generate standard curves, and amplify a 160 bp and a 130 bp fragment, respectively. All primer sequences are listed in S6 Table. Primer efficiencies, defined as the fraction of target molecules that are amplified in one PCR cycle, were 102.4 and 90.1% for *recA* and *res*, respectively.

To measure pConj copy number, bacteria were grown in 5ml GCBL until $OD_{600}$ 0.5–0.8. Two 10 μl samples were collected, washed in nuclease-free water, then resuspended in 100 μl nuclease-free water and boiled at 95˚C for 5 min. For qPCR, 1 μl of purified control plasmid or lysate was added to 19 μl of qPCR mastermix containing SYBR GREEN (Applied Biosystems), and 0.5 μM of each primer. Reactions were run on a QuantStudio Real-Time PCR System

(Thermofisher). Ct values were compared to standard curves as described previously [61], and the relative plasmid copy number ratio (P:C ratio) determined as the ratio of *res* to *recA*.

## Plasmid maintenance assays

Bacteria were grown overnight on solid media containing either kanamycin or tetracycline to ensure all colonies contain pConj, then used to inoculate 5 ml FB at $OD_{600}$ 0.01. Cultures were passaged every 20 hours as required to a final 30–40 generations. For shorter assays, two individual colonies were used to inoculate 5 ml FB and grown for 20 hours (6–9 generations). To detect plasmid loss, strains with pConj harbouring the *gfp:kan* cassette were serially diluted, plated, visualised with a UV transilluminator (Invitrogen), and the number of green and white colonies counted (approximately 300 colonies in total). In every experiment, all white colonies and 10–20 green colonies (chosen at random) were replica plated onto GCB plates containing tetracycline or kanamycin to confirm the absence/presence of pConj.

## Transposon insertion sequencing (TIS)

A library of *N. gonorrhoeae* FA1090 mutants was generated by *in vitro* mutagenesis. pHsk1 was constructed by Gibson assembly to contain the kanamycin resistance gene flanked by Hsmar inverted repeats, with backbone of pCE005 (gift from Dr Cara Ellison) carrying erythromycin resistance. In each 100 μl transposition reaction, 30 μg genomic DNA was subjected to mutagenesis with 125 nM Hsmar transposase (gift from Professor Ronald Chalmers) with 5 μg pHsk1 in reaction buffer (20 mM Tris-HCl pH 8, 100 mM NaCl, 10% v/v glycerol, 2 mM DTT, 2.5 mM $MgCl_2$) for 18 hours at 30°C. Products of eight transposition reactions were pooled, precipitated by the addition of 10 μl 3 M sodium acetate pH 5.2 and 800 μl isopropanol, resuspended in $ddH_2O$, and repaired in T4 DNA ligase buffer (50 mM NaCl, 0.5mM dNTPs) with 50 U T4 DNA polymerase (Thermofisher) and 10 U T4 DNA ligase (Invitrogen) at room temperature for 90 min. The repaired DNA (375 ng in 25 μl) was used to transform *N. gonorrhoeae* in 81 independent transformations.

The mutant library was used as the recipient in conjugation with FA1090 containing pConj. The library and donor were grown to $OD_{600}$ 0.6–0.8 in 20 ml GCBL, mixed at a 1:1 ratio, and added as 10 μl spots onto GCB plates before incubating for 24 hours. A total of 72 conjugations were performed. Transconjugants were harvested from plates containing kanamycin and tetracycline, and grown in 15 ml GCBL in duplicate, with or without tetracycline, for 20 hours starting at an $OD_{600}$ ~ 0.3. Every 12 hours bacteria were subcultured at $OD_{600}$ 0.1 in 15 ml FB with kanamycin +/- tetracycline. In total, bacteria were passaged 14 times (approx. 56 generations) over 7 days; the tetracycline concentration was increased to 10 μg/ml for the last passage. Genomic DNA from 3 x $10^9$ bacteria from each condition was extracted using Wizard Genomic DNA Purification kit (Promega, USA) every 24 hours. Nucleotide sequences were determined for genomic DNA from the initial library, 2nd, 4th and 14th passage (generations 0, 8, 16, 56) using the TraDIS-*Xpress* sequencing method [33].

For sequencing library preparation, the Nextera Tagmentation Enzyme kit (Illumina) was used. Custom oligonucleotides (S6 Table) were used instead of i5 index primers with 28 PCR cycles to add sequencing primer sequences and indexes [33,62]. Fragment sizes were checked using a 4200 TapeStation System (Agilent). The resulting DNA was purified using AMPure XP beads (Beckman Coulter), then sent for paired-end sequencing on a NextSeq500 machine (Illumina) using a High Output PE 150 (150 cycles), mixed with standard whole genome sequencing (WGS) shotgun bacterial libraries at Quadram Institute (Norwich, UK).

For plasmid stability assays with mutants identified in TIS, mutants were constructed using the relevant primers (S6 Table); specifically, approximately 1 kb upstream and downstream of

*neis1066* was amplified with 490/528 and 493/529 respectively. Primers 524/525 and 526/527 were used to amplify regions upstream and downstream of *neis1845*, while primers 530/531 and 532/533 were used to amplify upstream and downstream fragments around *neis2592*. The erythromycin resistance gene (*ermC*) was amplified with primers EryF/EryR. The final constructs were amplified with primers 490/493 (*neis1066*), 524/527 (*neis1845*) and 530/533 (*neis2592*). The construct for deleting *pilD* was amplified from FA1090Δ*pilD*::*ery* (gift from Tabea Elsener) with primers pilD_upF and pilD_downR. All constructs were introduced into FA1090 pC1$^{GFP}$.

For the stability assays, bacteria were grown overnight on solid media containing erythromycin and tetracycline then harvested and inoculated into 5 ml FB + 1% vitox at an $OD_{600}$ 0.1. Cultures were passaged every 12 hours for 16 generations. Plasmid loss was detected as previously by the detection of GFP visualised with a UV transilluminator. As all colonies were green, 20 green colonies (chosen at random) were replica plated onto GCB plates containing tetracycline or GCB to confirm the presence of pConj.

## Toxicity assays

To construct plasmids for toxicity assays, pBAD33 plasmid backbone was amplified with primers 357/358, and joined with *vapD*$^{pConj}$ (amplified with primers 359/360) using NEB HiFi assembly. For pBAD-*vapD*$^{pConj}$*vapX*$^{pCryp}$ and pBAD-*vapD*$^{pConj}$*vapX*$^{Nm}$, *vapD*$^{pConj}$ was amplified using primers 359/349, *vapX*$^{pCryp}$ (allele 1 in PubMLST) was amplified with primers 350/361 and *vapX*$^{Nm}$ (allele 22 in PubMLST) was constructed using primers 374/375 and 376/361. Plasmids were transformed into *E. coli* DH5α and sequenced. Bacteria were grown overnight (22–24 hours) in glucose and chloramphenicol, then sub-cultured in 5 ml LB with 0.2% glucose (w/v) until the $OD_{600}$ reached ~0.1. Bacteria were then centrifuged at 4500 x *g* for 10 min, washed, then resuspended in pre-warmed 5 ml of LB with 0.2% L-arabinose (w/v), and incubated over 6 hours shaking at 180 rpm. Samples were taken at 0, 15, 30, 60, 180 and 360 min and plated on LB agar with chloramphenicol and glucose. All primers are listed in S6 Table.

## Minimal inhibitory concentration assays

Minimal inhibitory concentrations (MICs) were determined in accordance to previously published protocols [63]. Briefly, strains were grown overnight on GCB agar then resuspended in GCBL to $OD_{600}$ 1.0; $10^4$ CFU in 5 μl were spotted onto GCB plates containing the appropriate antibiotics at two-fold dilutions. Once spots were dried, plates were incubated for 20–24 hours before MIC determination.

## pConj curing

Strains carrying pCΔGL with *gfp:kan* were grown on non-selective GCB agar for 18–22 hours before starting liquid cultures at 5 x $10^4$ CFU/ml in 5ml FB containing 0.5 MIC of each antibiotic. Cultures were then left static for 48 hours as described previously [25], and mixed well before plating on GCB. Approximately 200 colonies were inspected for loss of plasmid using a UV transilluminator (Invitrogen), resulting in a limit of detection of 0.5%. All white colonies and 10–20 green colonies were picked for replica plating. Cultures were also serially diluted at the end of the experiment and 5 μl were spotted onto GCB plates for CFU determination.

## Phylogenetic analysis

Isolates (n = 126) representing all seven pConj variants (S2 Table) were randomly selected from gonococcal pConj-carrying strains on the PubMLST database [64]. The Tn*916*

transposon sequence was identified by first aligning the consensus sequence of all seven pConj variants; *tetM*-containing sequence and its transposon was removed from the analysis. For generation of *N. meningitidis* pConj phylogenetic tree, all isolates carrying pConj on a single contig (35 kb– 44 kb, n = 70) were selected. pConj sequences were aligned with the Multiple Alignment using Fast Fourier Transform (MAFFT) program on Galaxy (www.usegalaxy.org). RAXML-NG (version 1.1.0) [65] using the GTRGAMMA model of nucleotide substitution was used to generate a phylogenetic tree. The program was run thrice independently with the default settings until either convergence was reached, or 1000 replicates were completed. The resulting tree was then visualised on MEGA-X (version 11.0.11).

For visualisation of pConj dissemination within the gonococcal population, *N. gonorrhoeae* isolates from S1 Table were selected on PubMLST [64]. Genetic relationships between gono-coccal isolates were determined through the core genome multilocus sequence typing scheme (Ng_cgMLST) then visualised using a minimum spanning algorithm using Grapetree software [66] as done previously [9]. The presence of pConj in each isolate was determined through genes unique for this plasmid (*traM* and *trbC*) [9].

## Analysis of plasmid distribution

Data analysis involving pConj, pCryp and p*bla* carriage in all *Neisseria* spp. populations was carried out using isolate sequences available on the PubMLST database [64]. All isolates were confirmed to have a sequence bin > 1 Mb and contigs ≤ 200 to ensure that sequences available were complete and well sequenced. For UK and US analyses, isolates prior to 2000s were not analysed due to inadequate number of samples. Presence of *traM* or *trbC* denoted pConj carriage (as above), while presence of *mobC* or *repA* denoted pCryp presence. Presence of p*bla* was denoted by the $bla_{TEM}$ gene (*neis2357)*. BLAST (https://blast.ncbi.nlm.nih.gov/Blast.cgi) was used to analyse for presence of sequences similar to pConj in non-*Neisseria* species.

## Statistical analyses

Sequence alignments of ζ2/ζ3, VapD and VapX were done with Clustal Omega [67] and visu-alised on Jalview [68]. Gene annotations are consistent with NEIS annotations in PubMLST. Structural prediction of VapD was performed with AlphaFold [69] and visualised using Pymol (The PyMOL Molecular Graphics System, Version 1.2r3pre, Schrodinger, LLC.) [70]. Analysis of TIS data was performed using Bio-Tradis (version 1.4.1) [71] and subsequently Artemis [72].

Statistical significance using either unpaired *t*-test, one-way ANOVA and two-way ANOVA with multiple comparisons if required (as indicated in figure legends) and normality tests for assays were performed with GraphPad PRISM v 9.0. Odds ratio was also similarly cal-culated using default parameters on PRISM v 9.0.

## Supporting information

**S1 Fig. Alignment of ζ3 and ζ2 amino acid sequences.** The two most prevalent protein vari-ants of each ζ3 and ζ2 (with their respective *neis2213* alleles in brackets) were used for the alignment. Two other characterised *Streptococcus pneumoniae* ζ proteins were included in the alignment (PDB numbers, 1GVN and 2P5T, are in brackets). Shading intensity is proportion-ate to percentage conservation. All ζ protein variants have conserved active site residues (cata-lytic residue and the corresponding amino acid on 1GVN is indicated by the arrow).
(TIF)

**S2 Fig.** *N. gonorrhoeae* **pConj is conserved compared to pConj in** *N. meningitidis.* Phylogenetic tree of *N. meningitidis* and *N. gonorrhoeae* pConj drawn separately but at the same scale. Each dot represents an isolate, colour-coded according to year of its isolation.
(TIF)

**S3 Fig. pConj is a low copy plasmid that does not impose a fitness cost.** (**A**) *gfp:kan* cassette on pConj does not affect growth rates. FA1090 carrying the respective plasmids were first grown overnight before liquid culture was set up using gonococcal base media (GCBL) and monitored over 8 hours. (**B**) No pConj loss was observed with FA1090 carrying pC1$^{GFP}$ or pC4$^{GFP}$. FA1090 +/- pConj grows at the same rate in (**C**) fastidious broth and (**D**) GCBL over 24 and 8 hours respectively. (**E**) Ratio of pConj to chromosome is 4.60 ±1.63, as determined by qPCR of *res* in pConj and *recA* on the chromosome using bacteria grown to mid-log phase. Results of three independent experiments were analysed with two-way ANOVA with Sidak's multiple comparisons and shown as mean ± SD. ns, $p > 0.05$.
(TIF)

**S4 Fig. No additional genes were identified to be involved in plasmid maintenance.** (**A**) Insertions in the control (kanamycin) and +Tet (kanamycin+tetracycline) conditions were similar in the 56th generation. (**B**) The profiles of 16 genes with more than one unique insertion, and with functions associated with other genes previously identified to be involved in plasmid maintenance in other bacteria, are shown here, after 16 generations of growth; gene orientation is shown. Gene annotations are consistent with NEIS annotations in PubMLST. No significant differences between control and test conditions were observed.
(TIF)

**S5 Fig. Bacterial survival in the presence of 0.5 MIC ciprofloxacin and spectinomycin are comparable in both strains.** Bacteria were grown overnight on GCB before incubation in 0.5 MIC of ciprofloxacin or spectinomycin for A) FA1090 and B) 60755. Cultures were serially diluted and 5 μl was spotted on GCB agar to determine CFU at the end of the experiment.
(TIF)

**S1 Dataset. Bio-Tradis analysis of transposon insertion sites of all libraries analysed.** Available to view or download from https://figshare.com/s/d61951edde6cf3d4128a
(ZIP)

**S1 Table. List of all Neisseria spp. isolates used for analysis.** Available to view or download from https://figshare.com/s/2cddeada71ba824e2849
(XLSX)

**S2 Table. List of 126 gonococcal isolates used to analyse pConj.** Available to view or download from https://figshare.com/s/759d74cd19c31c597110
(XLSX)

**S3 Table. List of N. gonorrhoeae isolates from UK and USA obtained from PubMLST.** Available to view or download from https://figshare.com/s/23bed34933a306c392e1
(XLSX)

**S4 Table. List of statistical output from Bio-Tradis for all genes from Bio-Tradis at passages (A) 2 and (B) 4 comparing the +Tet and control conditions.** LogFC refers to log fold change of the +Tet condition compared to the control condition, LogCPM refers to the logarithmic count per million reads, while the *q* value refers to the corrected *p* value. Available to view or download from https://figshare.com/s/94a4394060ce5d13a07a
(XLSX)

**S5 Table. List of strains used in this study.** Available to view or download from https://figshare.com/s/d2dbf0d466e893028bcf
(DOCX)

**S6 Table. List of primers used in this study.** Available to view or download from https://figshare.com/s/32dd0e09c61c1d2a403f
(XLSX)

## Author Contributions

**Conceptualization:** Wearn-Xin Yee, Muhammad Yasir, Ana Cehovin, Christoph M. Tang.

**Data curation:** Wearn-Xin Yee, Muhammad Yasir, A. Keith Turner, David J. Baker, Ana Cehovin, Christoph M. Tang.

**Formal analysis:** Wearn-Xin Yee, Ana Cehovin, Christoph M. Tang.

**Funding acquisition:** Wearn-Xin Yee, Christoph M. Tang.

**Investigation:** Wearn-Xin Yee, Ana Cehovin, Christoph M. Tang.

**Methodology:** Wearn-Xin Yee, Muhammad Yasir, A. Keith Turner, David J. Baker, Ana Cehovin, Christoph M. Tang.

**Project administration:** Ana Cehovin, Christoph M. Tang.

**Resources:** Christoph M. Tang.

**Software:** Muhammad Yasir, A. Keith Turner, David J. Baker, Ana Cehovin.

**Supervision:** Ana Cehovin, Christoph M. Tang.

**Validation:** Ana Cehovin, Christoph M. Tang.

**Visualization:** Christoph M. Tang.

**Writing – original draft:** Wearn-Xin Yee.

**Writing – review & editing:** Ana Cehovin, Christoph M. Tang.

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
