## [Decision Letter · Decision Letter 0]

26 Feb 2023

Dear Dr Tang,

Thank you very much for submitting your Research Article entitled 'Evolution, persistence, and host adaption of a gonococcal AMR plasmid that emerged in the pre-antibiotic era' to PLOS Genetics.

The manuscript was fully evaluated at the editorial level and by three independent peer reviewers. The reviewers appreciated the attention to an important problem, but raised some substantial concerns about the current manuscript. Based on the reviews, we will not be able to accept this version of the manuscript, but we would be willing to consider a revised version. We cannot, of course, promise publication at that time.

If you decide to revise the manuscript for further consideration at PLOS Genetics, please aim to resubmit within the next 60 days, unless it will take extra time to address the concerns of the reviewers, in which case we would appreciate an expected resubmission date by email to plosgenetics@plos.org.

We are sorry that we cannot be more positive about your manuscript at this stage. Please do not hesitate to contact us if you have any concerns or questions.

Yours sincerely,

Xavier Didelot

Academic Editor

PLOS Genetics

Lotte Søgaard-Andersen

Section Editor

PLOS Genetics

Reviewer's Responses to Questions

**Comments to the Authors:**

Reviewer #1: The paper "Evolution, persistence, and host adaption of a gonococcal AMR plasmid that emerged in the pre-antibiotic era" by Yee et al investigates the plasmid pConj which is widely distributed in N. gonorrhoeae and also present at a lower frequency in some other Neisseria spp. The study demonstrates that the markerless portion of pConj ir remarkably conserved over an 80 year period. At some point a lineage of the plasmid acquired a tetM gene conferring tetracycline resistance. Interestingly, the authors demonstrate that cessation of TET usage did not result in loss of the tetM-carrying plasmid at the level of gonococcal populations. They also demonstrate that pConj, which carries a split toxin-antitoxin system (TA), specifically the toxin component vapD, is maintained cooperatively with one or more antitoxin genes carried on the pCryp plasmid or on chromosomal islands. The authors also demonstrate that the plasmid imposes no detacable fitness-cost, despite representing as much as 10% of the total DNA content of carrying cells (the latter estimates takes copy number into account). Finally, it is shown that FLQ treatment can be used to clear the plasmid. The latter is particularly important in light of the fact that pConj plays a role in the disseemination of a third plasmid, pbla, which encodes a beta-lactamase only two nucleotide substitutions away from a full ESBL phenotype. As third generation cephalosporins are the cornerstone of current treatment schemes, a scenario of pbla evolving to carry an ESBL would be very serious, and the current study reveals critical insights which could be very useful in such a scenario.

Taken together, I think the paper is of significant importance, super-interesting, consise, and to the point. I would like to commend the authors for their work and only have minor comments and suggestions for improvement, which are listed below.

Author summary: I think lines 56-58 might be a bit opaque to the casual reader. Also, I think the authors should make clear when they state that " ... no plasmid loss detected" on line 55, what this means specifically, as it is stated towards the bottom that pConj loss was observed following the introduction of FLQ treatment at the population level.

Introduction: I really loved the first page of the intro for being extremely concises and informative. It contextualizes the study very well.

Results (and methods): On page 8, the authors describe the distribution on pConj in Neisseria spp. I dont think the methods section describes this part very well. Was pubMLST used for this as well?

Results (and methods): I think the authors could spend a few sentences backing up the claim that mobC+repA are 100% diagnostic of pConj presence. Is there any uncertainty around how these genes are detected/annotated on pubMLST?

Fig2 and associated text in the paper: Perhaps the authors could consider whether any bias in sequencing efforts could affect the estimated fraction of tetM+ pConj over time? I know from painful experience that gonococcal genome sequences are not generated in a completely random fashion and resistant isolates are often overrepresented. TetR might not generate too much fuzz these days, but could tetM+ pConj be associated with resistance to other drugs such as azithromycin or 3rd gen cephalosporins? Could this affect estimates? I dont think there are any easy answers to this, but it might be worthwhile giving this some thought.

Finally, I think it would be good if the authors could share metadata on presence/absence of plasmids and gene elements across Neisseria species, in the form of a csv file or similar.

Reviewer #2: Summary

Two plasmids found in many Neisseria gonorrhoeae isolates, pbla and pConj, were reported to carry antibiotic resistance cassettes leading to the discontinuation of penicillin and tetracycline for gonococcal treatment. This study describes the evolution and maintenance of the pConj plasmid, a narrow host plasmid found in two major forms. Originally it was markerless but later isolates carry a TetR gene. The study tracked the evolution of pConj when it was first identified, 80 years ago, before antibiotics were available.

They report that pConj, is well-conserved by analyzing the whole genome sequences in the PubMLST database of strains isolated from 1928 through the present. There were minor additions and deletions of genetic information, but they suggest that there were no gross alterations of the plasmid sequence. The lineage with a tetracycline resistance gene persisted even after tetracycline was no longer recommended as a treatment in the US and UK. They showed that carrying Conj doesn’t imply a modified fitness cost by using a fluorescent tag and was stably inherited independently of any selection that would occur with the use of tetracycline. Using a transposon insertion sequencing approach, they conclude that pConj doesn’t rely on any non-essential chromosomal gene for its maintenance. However, they discover that a toxin-antitoxin (TA) system is one way that pConj is maintained in the population. A ParAB partitioning system is also involved in stable plasmid maintenance. Finally, they showed that using ciprofloxacin can negatively alter pConj maintenance, which might explain the existence of plasmid-less strains.

Critique:

The report’s premise is based on the bioinformatic analysis of a subset of 129 isolates from a geographical distance of sites and times of isolation. Some potentially interestingly experimental analysis follows up this bioinformatic analysis. There are many appropriate controls and appropriate statistical tests. The data provides a new analysis of this plasmid and has the potential to direct public health policies. While some of the conclusions follow from the data, some data may be over-interpreted, there are no new concepts developed, and some experimental details are missing.

1. Whether the plasmid is conserved as stated is hard to discern from the data presented. Can the changes be represented graphically (Figure 1) to present the reader with a way to judge the conservation level? Are the other plasmids of similar size or chromosomal loci that can be compared to determine whether there is, or is not, a high level of conservation?

2. The authors provide no analysis to model how the tetracycline resistance determinant was acquired or other plasmid changes.

3. The analysis of plasmid stability in the TnSeq approach is confusing and possibly incomplete. Why was the representation of transposon insertions in the population decreasing over generations? This reduction in the insertion representation could be due to the growth differences of different mutants. Still, it is difficult to believe this would not occur in the first few generations as opposed to over so many generations. These data raise the question of whether the population was sampled properly. Finally, since this is a conjugative plasmid, the role of conjugal transfer in repopulating the cells that lost the plasmid was not considered.

4. The conclusion that pConj remains in the population after the use of tetracycline ceased is an important conclusion. It would be worth clarifying the timelines of isolation and parameters of tetracycline, especially since two countries with different health recommendations are involved as is some countries where antibiotics may not be regulated. Perhaps adding a timeline under Figure XX would clarify the antibiotic history and put the US, UK, and other isolates in perspective. Is it clear that there was no longer tetracycline use after the recommendations changed?

5. Figure 3 presents negative data and could be moved to the supplement. The conclusion that some identified chromosomal genes are false positives is premature. This conclusion should be supported by constructing the loss-of-function mutants and directly testing their role in plasmid maintenance.

6. The Meyer lab adapted the pConj-tet plasmid for the Hermes gene delivery system that can introduce genes into transformation noncompetent cells. This reference could be cited, perhaps in the introduction (Kupsch et al. Mol Gen Genet 250:558 1996).

Reviewer #3: The manuscript “Evolution, persistence, and host adaption of a gonococcal AMR plasmid that emerged in the pre-antibiotic era” by Yee and colleagues answer the question about why and how the pConj plasmid is maintained in the gonococcal population even after cessation of the use of tetracycline after first-line treatment. The authors show a very nice combination of computational and experimental analyses, for which they use appropriate methodologies and software. This is the first study up to my knowledge that tries to understand this, and I found it very interesting, as plasmids in N. gonorrhoeae are still an understudied area. The manuscript is very well written and can be followed without problems. I only have minor comments/suggestions for them:

Abstract

Line 33: “pConj confers tetracycline resistance”. It does when it carries the tetM gene, but some strains don’t. Would be better to say “pConj can confer tetracycline resistance”.

Results

Line 143: I think it is important to specify throughout the text that the dates of emergence of each plasmid are actually dependent on the collections we have, i.e. the first isolate from the collection with a pConj was from 1940, but it doesn’t mean that it was not present before. It is not present in isolates from 1928-1939 from the collection the authors are using but these old isolates are very scarce and all from a Danish study (Golparian et al, 2020).

Lines 143-149: I assume the authors state that Group I pConj appeared first because it includes the oldest isolates. However, how is the pConj tree in Fig1D rooted? From that tree, Group I pConj appears to emerge from Group II. It looks unrooted, I suggest rooting this tree so it is visually/phylogenetically more evident that Group I emerged first.

S1 Fig legend: please, specify what 1GVN stands for.

Lines 170-171: my understanding is that tetracyclines are not the first choice for uncomplicated gonorrhea, but doxycycline is used in certain (sometimes quite often) circumstances (doi: 10.1177/0956462420949126) as well as to treat other STIs. I do not think doxycycline has been completely absent from treatment since 2004. A background sporadic usage of tetracyclines may have allowed the tetM pConj plasmid to stay in the gonococcus population even after the removal of tetracyclines as first line treatment.

Line 308: the authors could also check if these results are consistent with the genomic data that they first use from PubMLST. They could easily detect mutations leading to ciprofloxacin resistance (in gyrA, parC – at least gyrA S91F) and assess whether these isolates carry pConj. This would be a way of re-confirming their results in a big clinical dataset.

**Have all data underlying the figures and results presented in the manuscript been provided?**

Reviewer #1: Yes

Reviewer #2: Yes

Reviewer #3: Yes

PLOS authors have the option to publish the peer review history of their article (what does this mean?). If published, this will include your full peer review and any attached files.

Reviewer #1: No

Reviewer #2: No

Reviewer #3: No

---

## [Editor Report · Decision Letter 1]

14 Apr 2023

Dear Dr Tang,

We are pleased to inform you that your manuscript entitled "Evolution, persistence, and host adaption of a gonococcal AMR plasmid that emerged in the pre-antibiotic era" has been editorially accepted for publication in PLOS Genetics. Congratulations!

Yours sincerely,

Xavier Didelot

Academic Editor

PLOS Genetics

Lotte Søgaard-Andersen

Section Editor

PLOS Genetics

Comments from the reviewers (if applicable):

**Data Deposition**

http://datadryad.org/submit?journalID=pgenetics&manu=PGENETICS-D-23-00082R1

**Press Queries**

---

## [Editor Report · Acceptance letter]

9 May 2023

PGENETICS-D-23-00082R1 

Evolution, persistence, and host adaption of a gonococcal AMR plasmid that emerged in the pre-antibiotic era 

Dear Dr Tang, 

We are pleased to inform you that your manuscript entitled "Evolution, persistence, and host adaption of a gonococcal AMR plasmid that emerged in the pre-antibiotic era" has been formally accepted for publication in PLOS Genetics! Your manuscript is now with our production department and you will be notified of the publication date in due course.

With kind regards,

Anita Estes

PLOS Genetics

On behalf of:
